# Flexible Coatings Facilitate pH-Targeted Drug Release via Self-Unfolding Foils: Applications for Oral Drug Delivery

**DOI:** 10.3390/pharmaceutics16010081

**Published:** 2024-01-06

**Authors:** Carmen Milián-Guimerá, Laura De Vittorio, Reece McCabe, Nuray Göncü, Samvrta Krishnan, Lasse Højlund Eklund Thamdrup, Anja Boisen, Mahdi Ghavami

**Affiliations:** The Danish National Research Foundation and Villum Foundation’s Center for Intelligent Drug Delivery and Sensing Using Microcontainers and Nanomechanics, Department of Health Technology, Technical University of Denmark, 2800 Kgs. Lyngby, Denmarklhth@dtu.dk (L.H.E.T.); mahgh@dtu.dk (M.G.)

**Keywords:** oral drug delivery, enteric release, flexible coating, elastomer devices

## Abstract

Ingestible self-configurable proximity-enabling devices have been developed as a non-invasive platform to improve the bioavailability of drug compounds via swellable or self-unfolding devices. Self-unfolding foils support unidirectional drug release in close proximity to the intestinal epithelium, the main drug absorption site following oral administration. The foils are loaded with a solid-state formulation containing the active pharmaceutical ingredient and then coated and rolled into enteric capsules. The coated lid must remain intact to ensure drug protection in the rolled state until targeted release in the small intestine after capsule disintegration. Despite promising results in previous studies, the deposition of an enteric top coating that remains intact after rolling is still challenging. In this study, we compare different mixtures of enteric polymers and a plasticizer, PEG 6000, as potential coating materials. We evaluate mechanical properties as well as drug protection and targeted release in gastric and intestinal media, respectively. Commercially available Eudragit^®^ FL30D-55 appears to be the most suitable material due to its high strain at failure and integrity after capsule fitting. In vitro studies of coated foils in gastric and intestinal media confirm successful pH-triggered drug release. This indicates the potential advantage of the selected material in the development of self-unfolding foils for oral drug delivery.

## 1. Introduction

The small intestine is the main absorption site for drug compounds following oral administration. Additionally, targeted drug release in this area enhances the therapeutic efficacy of most commercialized oral therapies [1,2]. Many active pharmaceutical ingredients (APIs) are sensitive to low pH levels and the presence of enzymes, leading to degradation in the stomach [3,4]. In this regard, enteric coatings play a critical role in oral dosage forms by offering a protective barrier against the harsh gastric environment and enabling targeted drug delivery to the small intestine [5]. Enteric coatings are used in traditional dosage forms such as tablets or capsules [6,7,8,9,10]. However, these formulations release the drug in the lumen in an omnidirectional manner, which entails a lower local concentration of drug molecules close to the intestinal wall. Therefore, only a part of the dose will be available for drug absorption [11]. 

Self-configurable proximity-enabling devices (SPEDs) have been developed as a non-invasive platform to systemically deliver poorly permeable drug compounds. This concept has been used to enable intimate contact and apply mechanical pressure on the epithelium via swellable or self-unfolding devices, thereby enhancing the transport and bioavailability of the API. Juginger and co-workers developed superporous hydrogels that incorporate drug-loaded cores for tailored expansion in the small intestine [12,13,14,15,16,17]. Despite the fact that these systems offer an omnidirectional release of the loaded drug compounds, they revealed an enhancement in the transport of different drugs through Caco-2 monolayers [12] and gastrointestinal (GI) tissue [13]. Using superporous hydrogels, the group obtained an insulin bioavailability of 1–2% compared to subcutaneous injection [16]. This is a promising result considering that the current state-of-the-art dosage forms for successful oral peptide delivery provide around 1% bioavailability when delivered as a standard oral tablet utilizing salcaprozate sodium (SNAC) as a permeation enhancer [18,19]. Nevertheless, it seems that this concept has not made significant progress over the past two decades, possibly indicating challenges associated with translating promising scientific results into highly loaded commercial dosage forms feasible for mass production. Recently, more invasive biomedical devices that allow for direct injections in the epithelium have shown promising results for the oral drug delivery of poorly permeable drug compounds. Impressive values of 10% and 70% bioavailability of biologics were obtained using a Luminal-Unfolding Microneedle Injector (LUMI) [20] and a Self-Orienting Millimeter-scale Applicator (SOMA) [21,22], respectively. Similarly, Rani Therapeutics developed RaniPill^®^, a self-inflatable balloon contained in an enteric capsule that delivers drugs via an epithelial injection in the small intestine, thereby obtaining a bioavailability comparable to subcutaneous injection [23,24,25]. Despite the impressive results, further investigation is required to determine the long-term effects caused by GI injections. In addition, the complexity of these devices may limit their compatibility with mass production, which ultimately increases the price of the final product.

In our group, we have focused on SPED concepts for oral drug delivery and developed injection-free devices that enable unidirectional release in close proximity to the intestinal epithelium. We hypothesized that unidirectional release formulations increase the local concentration of drug molecules near the epithelium. Higher plasma concentrations have been observed compared to multidirectional release dosage forms, which can potentially reduce the dose of drug compounds [11]. In addition, we have proven that the proximity between the absorptive barrier and such unidirectional release devices plays an essential role, presenting a 50% decrease in drug absorption for every distance increase of 130 µm from a Caco-2 cell monolayer [26]. Moving from our empirical knowledge to developing ingestible devices, Jørgensen et al. designed and produced self-unfolding foils (SUFs) for unidirectional release in close proximity to the intestinal wall [27]. The design comprises an elastomeric SUF with microcavities that are loaded with a solid-state drug compound and then coated and rolled into an enteric capsule. Upon the disintegration of the capsule in the desired location of the GI tract, the SUF unfolds and enables unidirectional release in close proximity to the epithelium. The success of this strategy relies on the elastomeric properties of the foil, which is composed of a biocompatible material, polydimethylsiloxane (PDMS). In our first proof-of-concept study, a modest relative bioavailability of 0.12% was observed in rats compared to subcutaneous injections [27]. Recently, Ghavami et al. further explored and enhanced crucial features of SUFs, obtaining a 15-fold increase in bioavailability compared to the previous study [28]. A substantial effort was made to implement surface treatments of PDMS foils to ensure better adhesion between the foil and the top coating. However, the deposition of suitable enteric materials on loaded SUFs which remain intact after rolling is still challenging. Cracks were observed in the top coating when fitting the foils into capsules. Hence, a flexible, immediate-release coating was used instead, and the dosage forms were administered rectally to avoid gastric exposure when performing pharmacokinetic studies in rats [28]. This demonstrates that further studies on flexible enteric coatings that can be used for different SPED-type devices, such as SUFs, are still relevant.

The present study aims to develop different mixtures of enteric polymers and a plasticizer spray-coated on SUFs to evaluate the relation between the inclusion of the plasticizer, (i) the flexibility properties of the coating, and (ii) the drug release profile at gastric and intestinal pH levels before and after the application of mechanical forces (Figure 1). For this purpose, we prepared different mixtures comprising enteric polymers and an intermediate-molecular-weight solid-state plasticizer, polyethylenglycol (PEG) 6000, and evaluated the mechanical properties of the top coatings after spray coating. Additionally, the prepared mixtures were spray-coated on SUFs loaded with a spray-dried powder of a commonly used model drug, paracetamol [29,30,31]. The top coating was inspected before and after fitting the SUF into a capsule. Finally, pH-triggered release was evaluated in vitro in gastric and intestinal media. This was carried out on SUF devices prior to and after rolling to evaluate the potential impact of having strained the top coating during folding and capsule insertion.

## 2. Materials and Methods

### 2.1. Materials

A Sylgard^TM^ 184 silicone elastomer kit was bought from Dow Chemical (Midland, MI, USA). Eudragit^®^ L100-55 (EL100-55) and Eudragit^®^ FL30D-55 (EFL) were both obtained from Evonik (Essen, Germany), whereas Kollicoat^®^ MAE 100-55 (KMAE100-55) was acquired from BASF (Ludwigshafen, Germany). Polyethylenglycol (PEG) 6000 (M_W_ 7000–9000 g/mol), which presents a polydispersity of 1 and Mw 6019 g/mol according to the literature, was acquired from Merck (Darmstadt, Germany) [32]. Paracetamol (98–102% purity) and poly(vinyl alcohol) (98–99% hydrolyzed), as well as PBS tablets, were purchased from Sigma-Aldrich (St. Louis, MO, USA). Isopropanol (IPA) and acetone were purchased from VWR International (Radnor, PA, USA). Ultrapure water used throughout the studies was obtained from a Q-POD dispenser from Merk Millipore (Burlington, MA, USA).

### 2.2. Fabrication of PDMS Foils

SUFs were produced in polydimethylsiloxane (PDMS), a material recognized for its chemical stability and commonly used in biomedical applications [33,34,35]. For that purpose, SUFs were fabricated as described in the literature [27,28] by casting a SYLGARD^TM^ 184 kit, which is a two-component kit consisting of a PDMS base and a curing agent. The two compounds were mixed in a 10:1 ratio (*w*/*w*) and then degassed for 40 min in a desiccator to remove potential air bubbles. The obtained mixture was poured onto a silicon master fabricated via deep anisotropic etching as previously described [27]. The silicon master features interconnected trenches which, upon replication, form hexagonal concave compartments in the SUFs. The PDMS was spin-coated on the silicon master using a Laurell WS-650-23B spin coater (Laurell Technology Corporation, Lansdale, PA, USA) at 500 rpm (acceleration was set to 100 rpm/s) for 30 s and then cured at 37 °C overnight [28]. The PDMS replica was then peeled off of the silicon master and cut into foils. For the inspection of the coatings before and after fitting into a size 9h capsule, PDMS was cut into rectangular pieces measuring 7 × 5 mm^2^, which is a suitable size for targeting the small intestine in a rat model. Conversely, foils of 5 × 5 mm^2^ were created for in vitro drug release studies. To increase the surface free energy of the produced PDMS SUFs and the adhesion of the spray-coated enteric top layer, the PDMS foils were subjected to a two-step surface treatment. Initially, the surfaces were activated in a UV-ozone process, making the PDMS hydrophilic, and to stabilize the effect of this dry oxidation treatment, the foils were dipped in a solution of 1% (*w*/*v*) polyvinyl alcohol (PVA) in ultrapure water [28,36]. The surface of the SUFs was characterized via scanning electron microscopy (SEM) using a Hitachi TM3030 Plus tabletop microscope (Hitachi High-Technologies Europe, Krefeld, Germany).

### 2.3. Preparation and Incorporation of SUFs into a Capsule

Drug loading was carried out manually by pressing the spray-dried paracetamol powder into the compartments on the PDMS foil and then gently removing any excess powder as previously described [27]. To obtain a more uniform and smaller particle size and to facilitate the loading procedure, paracetamol was dissolved in ultrapure water and spray dried (Mini Spray Dryer B-290, Büchi, Flawil, Switzerland) using a high-performance cyclone and small-volume sample collector. The foils were weighed before and after loading to determine the total loaded amount of spray-dried paracetamol. The loaded SUFs were sealed with different mixtures of enteric materials that dissolve at a pH above 5.5 and a plasticizer using ultrasonic spray coating (ExactaCoat system, Sono-Tek, Milton, NY, USA). For this purpose, solutions of 2% (*w*/*v*) EL100-55 containing 0 (E0), 12.5 (E12.5), and 25% (E25) (*w*/*w* in relation to the polymer) PEG 6000 as a plasticizer were prepared in IPA. The same preparation method was used for solutions containing KMAE100-55 and 0 (K0), 12.5 (K12.5), and 25% (K25) PEG 6000. PEG 6000 was selected as a plasticizer due to its recommendation for use together with Eudragit^®^ and Kollicoat^®^ polymers at concentrations ranging up to 25% according to the literature [37,38]. In addition, a solution of 2% (*w*/*v*) EFL, a highly flexible polymer, was prepared in the same solvent. The spray coater was equipped with a microbore-fitted 120 kHz Vortex nozzle programmed to linearly move across the samples with a distance of 5 cm between the tip and the samples for a total of 800 passages [28]. During spray coating, the pump rate was kept at 0.5 mL/min and the generator power at 3 W, and the nozzle translation speed was set to 50 mm/s. The cladding air pressure and the hot plate temperature were set to 0.04 bar and 40 °C, respectively. Once loaded and sealed by the selected enteric polymers, the coating quality was inspected before and after the foils were rolled manually and inserted into size 9h gelatin capsules. The thickness of the respective top coatings was measured via contact profilometry (Alpha-Step IQ Stylus Profilometer, KLA-Tencor Corporation, Milpitas, CA, USA). For that purpose, different mixtures of the polymers were sprayed on flat silicon chips using the SUF spray-coating protocol outlined above. The coating thicknesses were measured by making a scratch in the coating and running measurements across it using a 3 mg tip force with a scan speed of 130 µm/s and a resolution of 0.4 µm. All measurements were performed with three replicates for each polymer mixture and repeated a total of three times in different locations on each sample. Visualization of the loaded and coated foils, before and after fitting into a capsule, was carried out via SEM. 

### 2.4. Tensile Characterization of the Coatings Using a Texture Analyzer

Mechanical characterization of the enteric coatings used for foil preparation was performed using a Texture Analyzer (TA.XTplusC Texture Analyzer, Stable Micro Systems, Godalming, UK). For that purpose, PDMS foils were produced as described in the section “Fabrication and Incorporation of SUFs into a Capsule” via spin coating on a flat silicon wafer instead of the hexagonal patterned silicon master. The dogbone samples used for the pull tests (gauge length, 26.7 mm, and gauge width, 7.3 mm) were obtained via CO_2_ laser cutting (Epilog Mini 24, 40 W system, Golden, CO, USA) with speed, laser power, and frequency settings of 25%, 15%, and 2500 Hz, respectively. The produced flat PDMS foils were then surface-treated and spray-coated with the different enteric polymeric mixtures, as described in the previous sections, and the total thickness was measured using a digital micrometer (Insize 3109-25). A 10 kg load cell and a A/TG screw-initiated vice clamp operating on knurled jaw faces (35 mm × 35 mm) were used to carry out the pull test at a constant rate of 1 mm/s. Each type of sample was measured with five replicates.

### 2.5. In Vitro Release Studies

In vitro drug release studies from the enteric-coated foils were carried out using a µDiss Profiler (Pion Inc., Billerica, MA, USA), as previously described in the literature [11,39,40,41]. The release was measured in PBS adjusted to pH levels of 2.5 and 7.5, which are similar to the gastric and intestinal pH levels in fasted rats, respectively [42]. To confirm that the foils ensure the protection of the encapsulated API at a gastric pH after its incorporation in a capsule, in vitro studies in PBS at a pH of 2.5 were repeated after the rolling of the final device. Initially, paracetamol calibration curves from 20 to 165 µg/mL were constructed through the addition of different volumes from stock solution in 10 mL of PBS at a pH of 2.5 or 7.5. In situ fiber-optic probes with a path length of 1 mm were applied, and the UV absorbance was measured in the range of 265–275 nm. For the release studies, the foils were attached to cylindrical magnets with double-sided carbon tape, transferred into a sample vial, and covered with 10 mL of media. UV measurements were carried out every 10–30 s over the course of 13 h in gastric conditions and until the total release of paracetamol was observed in the intestinal media. All studies were performed at 37 °C with a stirring rate of 100 rpm and with four replicates. 

### 2.6. Data Treatment

All data were treated using Microsoft Excel 2016 (Redmond, WA, USA) and expressed as mean ± standard deviation (SD) values unless stated otherwise. Where appropriate, a statistical analysis was carried out in GraphPad Prism version 9.5.0 (San Diego, CA, USA) using Student’s *t*-test. *p*-values below 5% (*p* < 0.05) were considered statistically significant. 

## 3. Results and Discussion

### 3.1. Preparation and Incorporation of SUFs into a Capsule

SUFs are designed to protect a loaded drug in gastric conditions while ensuring unidirectional drug release in the small intestine upon the dissolution of the enteric top coating. Figure 1 and Figure 2A show an overview of the preparation steps associated with the production of SUFs. PDMS SUFs with dimensions of 5 × 5 mm^2^ and 7 × 5 mm^2^ with an approximate thickness of 300 µm (Appendix A) were loaded manually with 1.01–1.74 mg (1.16 ± 0.11 mg, *n* = 56) and 1.81–2.12 mg (1.94 ± 0.10 mg, *n* = 8) of paracetamol, respectively. An SEM examination of the SUFs showed efficient and uniform loading of the cavities (Figure 2A). The foils were then sealed with different enteric polymer coatings in order to ensure that no loss of the loaded drug occurred during rolling and before targeted release. UV-ozone and PVA treatment ensured adhesion between the different coatings and the PDMS surface. The different polymeric mixtures were spray-coated with a total thickness of 25.75 ± 2.67 µm. The individual thickness values of the respective mixtures are reported in Appendix A. Inspection with SEM showed uniform coatings and successful covering of the drug-loaded compartments with the enteric polymer mixtures after spray coating (Figure 2B). 

After successful loading and coating, 7 × 5 mm^2^ SUFs were rolled up and fitted inside the body of 9h capsules. It was observed that EFL was the only coating that remained intact and uniform after foil rolling and capsule loading, thereby suggesting that this polymer is suitable for making SUFs that can be loaded inside capsules in a rolled-up conformation (Figure 2B(i)). When evaluating the integrity of the EL100-55 and KMAE100-55 mixtures, it was observed that the layers containing PEG 6000 presented fewer cracks and uncoated areas compared to the plasticizer-free coatings (Figure 2B(ii,iii)). This suggests that the presence of the plasticizer increases the flexibility and integrity of the coatings during rolling, as expected. However, all mixtures presented cracks or uncoated areas due to the detachment of the top coating. No major visual differences in the uniformity were observed between the coatings that contained 12.5% and 25% PEG 6000. 

### 3.2. Tensile Characterization of the Coatings Using a Texture Analyzer

Tensile characterization was used to evaluate the strain at failure and the stress–strain curves of the different enteric-spray-coated mixtures applied to the SUFs (Figure 3A). The total thickness of the coated dogbone-shaped PDMS was 479 ± 35 µm. For the final application, flexible coatings that can sustain large elastic and/or plastic deformations without being subject to ruptures are needed. As expected, according to the supplier specifications [43], the dogbone-shaped PDMS foils coated with EFL showed the highest strain at failure compared to the rest of the coated samples, presenting a value of 159.5 ± 23.4% (Figure 3B). When evaluating EL100-55 mixtures containing different proportions of plasticizer, it was observed that the inclusion of PEG 6000 increases the strain at failure, presenting values of 2.0 ± 0.5%, 2.8 ± 1.4%, and 116.8 ± 12.7% for E0, E12.5, and E25, respectively. A similar result was detected when KMAE100-55 was utilized, revealing values of 1.8 ± 0.3%, 3.4 ± 0.8, and 119.7 ± 15.2% for the mixtures K0, K12.5, and K25, respectively. No major difference was observed between the mixtures containing 0 and 12.5% PEG 6000, whereas a drastic increase was noted for mixtures containing 25% plasticizer (Figure 3B). Uncoated UV-ozone and PVA-treated PDMS samples were used as references and presented the highest values (241.5 ± 65.7%), as expected based on data from the literature [44]. The individual strain-at-failure graphics are shown in Appendix A. During tensile testing it was noticed that when the top coatings tore, the underlying PDMS also tore. Since it was observed that the spray-coated top coatings are less flexible than the underlying PDMS (Figure 3B), they can support higher stress without deforming significantly. Therefore, when the top coating fails, the total force is applied on the underlying PDMS, which is then rapidly deformed and fails.

In Figure 3C, a cross-sectional SEM image of a foil residing in a size 9h capsule can be observed. The foil is w = 7 mm wide and l = 7 mm long and has a nominal approximate thickness of t = 300 µm (Appendix A). The SUF presents an arithmetic spiral conformation, and we can assume that the strain ε is zero in a center plane located in the middle of the bulk foil. Based on geometrical considerations, the numerical strain can be evaluated at discrete locations corresponding to the radius of the center plane (r_c_) for the respective windings [28]. This entails that ε = t/(2r_c_), and it results in strain values of 0.18 and 0.31 for the outer and inner windings, respectively (Figure 3C). Based on this, the spray-coated top coatings should present a strain-at-failure value higher than 31% to be able to support rolling and capsule loading without breaking. Therefore, theoretically, only EFL and the mixtures containing 25% plasticizer, E25 and K25, should remain intact after rolling the foil. Nevertheless, only EFL remained intact after fitting into a capsule (Figure 2B(i,ii)). The detachment of E25 and K25 from the rolled foil could be explained by the different adhesion properties of the mixtures to the PDMS-treated foils.

### 3.3. In Vitro Release Studies

To investigate drug release profiles in simulated gastric and intestinal conditions in fasted rats, the in vitro release of spray-dried paracetamol from SUFs was evaluated using a µDiss Profiler^TM^ (Figure 4A). In this study, we aim to (i) compare drug release profiles between SUFs coated with different mixtures of enteric polymers and a plasticizer, as well as to (ii) compare the release profiles in an acidic pH before and after rolling (and unrolling) SUFs coated with EFL, the only enteric coating in which no cracks were observed when fitting into the capsule (Figure 2B). 

During the studies performed in gastric conditions, no difference was observed in the drug release profile of the SUFs coated with EFL before and after rolling (31.4 ± 9.7% and 30.8 ± 8.0% after 2 h, respectively) (Figure 4B(ii–iv)). In addition, both situations presented approximately 75% drug release after 13 h. From this result, it could be concluded that the mechanical forces applied during the rolling of the foil did not affect the release properties of the loaded SUFs. This correlates with the SEM characterization of the coating before and after fitting in a capsule, where no cracks were observed (Figure 2B(i)). This is also linked with the tensile test in which the EFL coating presented one of the highest strains at failure (Figure 3B). When observing the drug release profiles in gastric conditions of SUFs coated with EL100-55, it can be seen that the inclusion of a plasticizer increases the release rate of paracetamol, which was 29.4 ± 5.7%, 43.2 ± 20.1%, and 69.1 ± 13.8% after 2 h for E0, E12.5, and E25, respectively (Figure 4B(ii,iv)). This can be explained because of the high aqueous solubility of PEG 6000 [45], which is independent of pH, as well as its disintegrant properties in solid dosage forms to facilitate drug release [46]. For this application, the different mixtures of KMAE100-55 did not reveal successful drug protection (60–70% drug release after 2 h), regardless of the amount of plasticizer used (Figure 4B(i,iv)). It was also noticed that all samples presented an early release of paracetamol (at least 30%) after 2 h in simulated gastric conditions (Figure 4B(iv)). A similar premature drug release after exposure to a medium with a low pH has been detected in previous studies [40,41,47]. To confirm that the premature release was not simply the result of a dissolved coating layer, the SUFs were inspected using SEM after the studies at a gastric pH. As expected, the coating layer remained on the top of the SUFs. Therefore, we believe that the premature release could be caused by the small molecule size and the high water solubility of paracetamol, which may allow for the diffusion of the drug through the enteric coating layer [48]. 

For the studies performed at an intestinal pH, a total release of paracetamol was successfully attained from SUFs coated with the different polymeric mixtures (Figure 4C(i,ii)). No differences in the release profiles were observed between the mixtures containing EL100-55 and KMAE100-55, for which a total release of paracetamol was achieved after approximately 20 min. When evaluating the release profile of SUFs coated with EFL, a slower drug release was noticed (total release after approximately 80 min). This could be explained by the material properties of EFL, which is a combination of Eudragit^®^ L30D-55, an enteric polymer, and a flexible sustained-release polymer, Eudragit^®^ NM 30D [43]. It is believed that the presence of a sustained-release material that disaggregates over time may delay drug release in comparison with fully enteric polymers that readily dissolve at a pH above 5.5, such as EL100-55 and KMAE100-55. A slightly slower release at an intestinal pH could be preferred due to the time required for the capsule to completely disintegrate. Gelatin capsules become adherent when dissolving, which can cause fouling and interfere with the release of the formulation to the tissue surface. Therefore, a more sustained release can be advantageous in preventing premature drug release until the complete opening of the SUFs.

## 4. Conclusions

Overall, we investigated flexible enteric coatings for SPED-type devices, such as SUFs, that enable the encapsulation and protection of solid-state formulations until arrival in the small intestine. Compared to previous work in this area, we evaluated (i) the integrity of enteric coatings applied on loaded SUFs, (ii) how the incorporation of PEG 6000 as a plasticizer affected the flexible properties of the spray-coated mixtures and their respective drug release in gastric and intestinal conditions, and (iii) the prevention of premature drug release before the SUF is fully opened at the site of delivery.

Among all the enteric coatings evaluated, EFL seems to be the most suitable material for flexible SPED devices, such as SUFs. It presented the best performance in terms of sustaining deformations, the visual characterization of the coating, and in vitro protection under gastric pH conditions after applying mechanical forces. Our findings demonstrated that SUFs can be loaded, coated with EFL, and successfully fitted into a capsule without compromising the integrity of the layer. In addition, protection in a gastric environment with a maximum of 30% drug loss after 2 h was observed when evaluating SUFs coated with EFL, as similarly observed in previous studies [40,41,47]. It was also shown that while the addition of plasticizers, in this case PEG 6000, can improve the flexibility of coating materials, it potentially compromises their enteric properties. The current studies were conducted using a highly soluble and commonly used model drug, paracetamol [29,31]. However, further investigations involving macromolecules, such as insulin, are required to fully understand how drug-specific properties contribute to the coating integrity and drug protection of EFL, the most suitable enteric coating for SUFs among all the materials evaluated in this study [49].

## Figures and Tables

**Figure 1 pharmaceutics-16-00081-f001:**
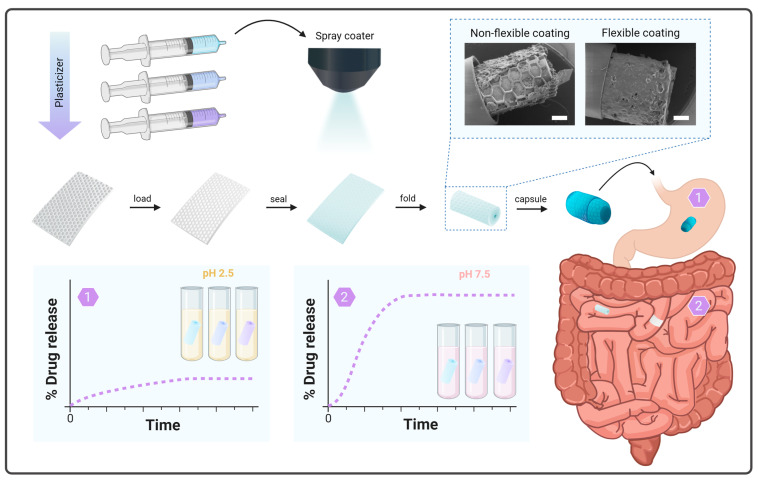
Schematic overview of the preparation steps and functionality of SUF devices utilizing different mixtures of enteric polymer and plasticizer. SUFs are loaded, spray coated with different mixtures, rolled, and finally fitted inside a capsule. The capsule passes through the stomach and disintegrates in the intestine. Then, the SUF unfolds and releases the drug unidirectionally in close proximity to the epithelium. To investigate the enteric properties of the coatings, in vitro studies are conducted in gastric and intestinal media. All scale bars represent 500 µm.

**Figure 2 pharmaceutics-16-00081-f002:**
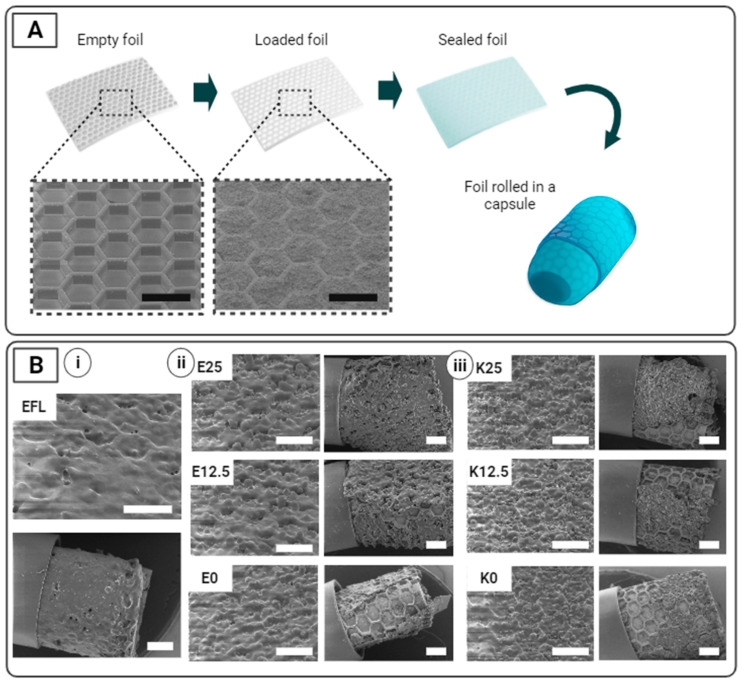
Drug loading and top coating before and after SUF rolling. (**A**) Schematic illustration the preparation steps of the final oral delivery device with SEM images of an empty and drug-loaded SUF. (**B**) SEM images of the (**i**) EFL-coated SUF before (**top**) and after (**bottom**) rolling; (**ii**) EL100-55-coated and (**iii**) KMAE-100-55-coated SUFs with different amounts of plasticizer (25, 12.5, and 0%) before (**left**) and after (**right**) rolling. All scale bars represent 500 µm.

**Figure 3 pharmaceutics-16-00081-f003:**
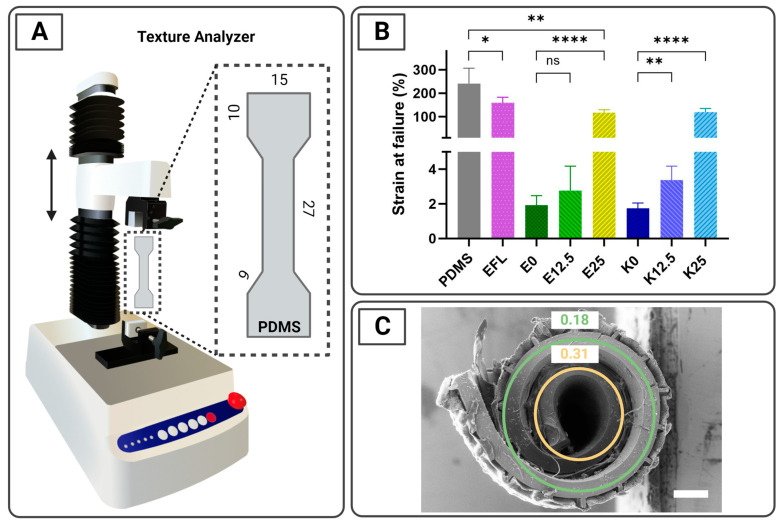
Tensile test studies. (**A**) Texture Analyzer setup to characterize the mechanical properties of flat PDMS dogbone samples. All dimensions are expressed in mm. (**B**) Bar graphics showing the strain at failure of top coatings spray-coated onto surface-treated PDMS dogbone samples. Uncoated PDMS was used as a reference. Data are shown as mean ± SD values (*n* = 4–5). Statistical significance is presented as * (*p* ≤ 0.05), ** (*p* ≤ 0.01) and **** (*p* ≤ 0.0001). ns indicates a not significant difference. (**C**) SEM image of a cross-section of a 300 µm thick foil residing in a 9h capsule (diameter = 2.69 mm). The circles show the approximate center plane of each winding, and the numbers correspond to the strain. The radii of the outer and inner center planes (r_c_) are 820 µm and 474 µm, respectively. The scale bar represents 500 µm.

**Figure 4 pharmaceutics-16-00081-f004:**
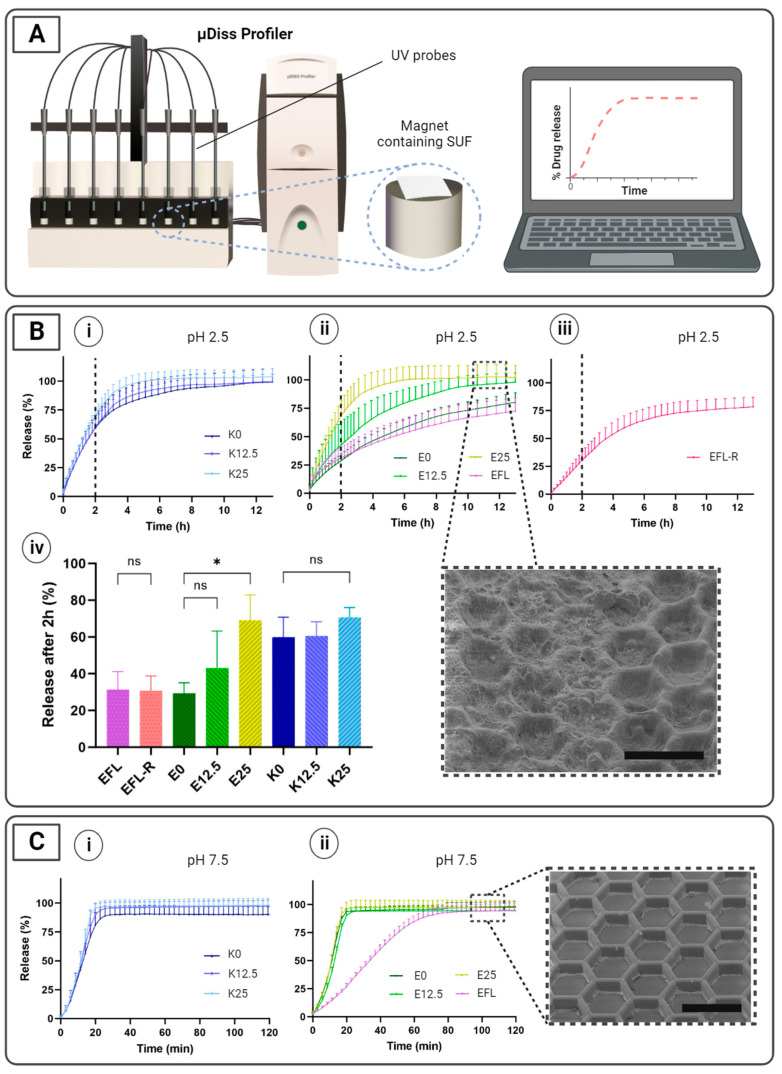
In vitro drug release studies. (**A**) µDiss Profiler setup applied to study the release profiles in vitro. Individual fiber optic probes allow for simultaneous measurements. SUFs were attached to a cylindrical magnet, and drug release was measured as the concentration of paracetamol in the solution over time. (**B**) In vitro release in PBS at a pH of 2.5 of paracetamol-loaded and (**i**) KMAE100-55-coated and (**ii**) EL100-55-coated samples with different concentrations of plasticizer (25, 12.5, and 0%) and EFL before and (**iii**) after rolling together with (**iv**) a bar graphic of paracetamol released after 2 h. The SEM image illustrates that the coatings appear intact after the release studies in an acidic pH. * indicates a significant difference with a *p*-value below 5%. Not significant difference is presented as ns. (**C**) In vitro release in PBS at a pH of 7.5 of paracetamol-loaded and (**i**) KMAE100-55-coated and (**ii**) EL100-55-coated samples with different concentrations of plasticizer (25, 12.5, and 0%) and EFL before rolling. The SEM image illustrates that the compartments were completely empty at the end of the release study at an intestinal pH. All scale bars represent 500 µm. Data are shown as mean ± SD values (*n* = 3–4).

## Data Availability

Data will be available on request.

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
