# Peer review of "Flexible Coatings Facilitate pH-Targeted Drug Release via Self-Unfolding Foils: Applications for Oral Drug Delivery"

_pharmaceutics, 2024, doi:10.3390/pharmaceutics16010081_

Round 1
Reviewer 1 Report
Comments and Suggestions for Authors
The manuscript investigates flexible enteric coatings for self-configurable proximity enabling devices that enable encapsulation 284 and protection of solid-state formulations until arrival to the small intestine.. The work is very interesting, and I believe its ready for publication after minor improvements.
1. The authors have already reported that the elf-unfolding foil (SUF) as an effective oral delivery device for macromolecules in the J. Contol. Relaese (https://doi.org/10.1016/j.jconrel.2023.07.041). However, in conclusion of this manuscript, the authors described as follows; However, further investigations involving macromolecules, such as insulin, are 297 required to fully understand how drug-specific properties contribute to coating integrity and drug protection
1. The authors never provide a citation for previously published article in this paper. What is the novelty of this manuscript comparing with previous author's result? Please try to emphasize the novelty of this manuscript.
2. In Fig. 4B ii, the shape change of the formulation surface is observed in the SEM image at 12 hours after the start of the dissolution test, and the evaluation of dissolution characteristics is based on the change at 2 hours after the start of the test. Do the authors have any observation results of the surface shape of the formulation at 2 hours after the start of the dissolution test? Please discuss the relationship between those observations and the dissolution rate.
3. The reviewer would like to add a legend to Figure 4C to make it easieras same as Figure 4B to decipher the different symbols. It is defined in the text, but it would be nice if it were incorporated into the figure itself.
Reviewer 2 Report
Comments and Suggestions for Authors
This is an interesting study about how flexible coatings facilitate pH-targeted drug release via self-unfolding foils. I suggest it for publication after the following minor points are well addressed.
1. The information about the polydispersity of PEG6000 should be added.
2. The resolution of Figure 3 should be improved to a higher level.
3. The authors should add more discussion about why PEG6000 was chosen to be used in this study.
4. Line 33-35, one recent study (https://doi.org/10.1016/j.cclet.2023.109029) should be included to support such a claim.
Comments on the Quality of English LanguageMinor editing of English language required
Reviewer 3 Report
Comments and Suggestions for Authors
The research work “Flexible Coatings Facilitate pH-targeted Drug Release via Self-unfolding Foils: Applications for Oral Drug Delivery” focuses to t developing different mixtures of enteric polymer and plasticizer spray coated on Self-unfolding Foils to evaluate the relation between inclusion of plasticizer, by improving the flexibility properties of the coating and the drug release profile at gastric and intestinal pH before and after application of mechanical forces. The work is novel and as per scope of journal.
The comments and observations are as follows:
1. Figure 1 should be consider as graphical abstract.
2. Why, activated in a UV-ozone is done?
3. Line 123: The drug loading was carried out manually by pressing the spray dried paracetamol powder into the compartments on the PDMS foil and then gently removing any excess powder as previously described.
If this is done manually…how one care assure uniformity.
4. If available, provide thickness data, before after coating. Also show data of percentage wait gain.
5. What is max and minimum dose that can be loaded in such foil?
6. What about stability of this foids?
